# Academic achievement after a CT examination toward the head in childhood: Follow up of a randomized controlled trial

**Elina Salonen**[1]*, **Robert Bujila**[2], **Jean-Luc af Geijerstam**[3], **Håkan Nyman**[4],
**Olof Flodmark**[4,5], **Peter Aspelin**[6], **Magnus Kaijser**[4,5]

1 Department of Medicine Solna, Karolinska Institutet, Stockholm, Sweden, 2 Medical Radiation Physics and Nuclear Medicine, Karolinska University Hospital, Stockholm, Sweden, 3 The Swedish Agency for Health and Care Services Analysis, Stockholm, Sweden, 4 Department of Clinical Neuroscience, Karolinska Institutet, Stockholm, Sweden, 5 Department of Neuroradiology, Karolinska University Hospital, Stockholm, Sweden, 6 Department of Clinical Science, Intervention, and Technology, Karolinska Institutet, Stockholm, Sweden

* elina.salonen@ki.se

**Data Availability Statement:** Data cannot be made freely available as they are subject to secrecy in accordance with the Swedish Public Access to

## Abstract

### Introduction

Increasing use of CT examinations has led to concerns of possible negative cognitive effects for children. The objective of this study is to examine if the ionizing radiation dose from a CT head scan at the age of 6–16 years affects academic performance and high school eligibility at the end of compulsory school.

### Materials and methods

A total of 832 children, 535 boys and 297 girls, from a previous trial where CT head scan was randomized on patients presenting with mild traumatic brain injury, were followed. Age at inclusion was 6–16 years (mean of 12.1), age at follow up 15–18 years (mean of 16.0), and time between injury and follow up one week up to 10 years (mean of 3.9). Participants' radiation exposure status was linked with the total grade score, grades in mathematics and the Swedish language, eligibility for high school at the end of compulsory school, previously measured GOSE-score, and their mothers' education level. The Chi-Square Test, Student's *t*-Test and factorial logistics were used to analyze data.

### Results

Although estimates of school grades and high school eligibility were generally higher for the unexposed, the results showed no statistically significant differences between the exposed and unexposed participants in any of the aforementioned variables.

### Conclusions

Any effect on high school eligibility and school grades from a CT head scan at the age of 6–16 years is too small to be detected in a study of more than 800 patients, half of whom were randomly assigned to CT head scan exposure.

Information and Secrecy Act. Because of this, the authors don't have the physical data set nor the legal right to share it. Instead, the authors have access to the data set through Statistics Sweden's MONA (a platform for access to microdata). Access to data can be made possible to researchers upon request (subject to a review of secrecy) in the same manner as for the authors, by linking OCTOPUS-data with the register data from Statistics Sweden. Requests for OCTOPUS-data should be made to Sven Oredsson, Sven. Oredsson@skane.se. Requests for linkage with the data registers presented in the article should be made to Statistics Sweden, mikrodata@scb.se.

**Funding:** M.K, H.N, O.F, J-L.G received a grant from a regional agreement on medical training and clinical research, ALF, between Stockholm County Council and Karolinska Institutet: "Påverkar strålningen från datortomografi barns kognitiva förmåga – uppföljning av en randomiserad klinisk prövning FoUI-946983". https://www. regionstockholm.se/om-regionstockholm/ Information-in-English1/ The funders had no role in study design, data collection and analysis, decision to publish, or preparation of the manuscript.

**Competing interests:** The authors have declared that no competing interests exist.

## Introduction

The number of computed tomography (CT) examinations has grown rapidly since its introduction in the 1970s [1, 2]. In the year 2018 the number of CT scans in Sweden was 1.5 million, an increase of 130% compared to 2005 [3]. Over time, technological advances and awareness of radiation protection has led to lowered doses to the head in diagnostic CT scans, but the comparably high doses of ionizing radiation from CT remain a concern, especially since the number of scans continue to increase [4–6].

Ionizing radiation can damage living cells by inducing strand breaks in DNA [7]. In some cases, this may lead to cancer, but most often it leads to cell death. Thanks to the regenerative capacity of tissue, this does not normally affect organ function. In the brain, however, neurons do not regenerate as in other organs, and the detrimental effects on the developing brain from high doses of ionizing radiation are well known. Previous research has assumed a threshold dose for cognitive effects at around 1–2 Gy for the developing child brain [8], whereas the effects from lower doses, such as the ones used in CT examinations, are less known. However, increased cancer risks have been reported for doses around 50–60 mGy to the head [9], which overlaps with the radiation dose from a single CT head examination [10]. Since cell death is a more common effect from ionizing radiation than cancer, this raises the question if doses in the range of a CT scan could have negative implication also for cognition [11].

In a study of young children irradiated for cutaneous hemangiomas, Hall et al. [12] showed that high school attendance decreased from 32% for non-exposed to 22% for exposed with 100–250 mGy. Furthermore, they found a negative dose-response relationship for high school attendance. However, a similar cohort study by Blomstrand et al. [13] could not find a difference in completion of post high school education for the exposed and unexposed. Problems with these results include confounding by indication, where the reason for radiation, rather than the radiation itself, could be the cause of the cognitive decline. A previous follow-up study of a clinical trial of management strategies after mild traumatic brain injury (mTBI) where exposure and non-exposure to ionizing radiation from a CT examination were randomized, no statistically significant differences between the exposure groups in a variety of cognitive measures such as motor speed, reaction time, selective attention, executive function, visuospatial ability, or memory were found [14]. However, the sample was small and, hence, the statistical power could be put in question.

The aim of the present study was to examine if the ionizing radiation dose from a CT examination of the head at the age of 6–16 years affects academic performance and high school eligibility at the end of compulsory school.

## Materials and methods

### Participants and procedure

The present study followed up participants from the OCTOPUS-study, a randomized controlled trial (RCT) comparing two management strategies for mild traumatic brain injury (mTBI)- inhospital observation or a CT examination and discharge (if normal examination findings) from the hospital [15]. The OCTOPUS-study included mTBI-patients over six years of age and mTBI was defined as head injury with amnesia and/or loss of consciousness, a Glasgow Coma Scale value of 15, and normal neurological findings. In the present study, inclusion was restricted to patients who were between 6 and 16 years old and who had not yet graduated from compulsory school at the time of their head injury. For this register linkage study consent was not informed, as the need for consent was waived by the ethics committee (Ethical Review Board of Stockholm at Karolinska Institutet, 2011/3:4). The risk with carrying out the linkage

study without informed consent is the possible violation of privacy. However, this must be weighed against the fact that the identities of the participants are unknown to us and consent would require that the identities be revealed. The benefit of the study is that possible risks with computed tomography can be studied. Since computed tomography is a very common examination method, it is important that its safety is evaluated. Since the risk of a breach of privacy can be considered small, we believe that the benefit of the study far exceeds its disadvantages.

From the OCTOPUS-study, information on the Personal Identification Number (PIN) including birthdate, date of inclusion/possible exposure to ionizing radiation, randomization allocation, exposure to ionizing radiation status, sex of participants, and Glasgow Coma Scale Extended (GOSE) score at 3 months after injury date was retrieved. With the PIN, radiation exposure doses could be retrieved or reconstructed through information in PACS at the Karolinska University Hospital. With the PIN, Statistics Sweden (SCB) linked the study subjects' data with the final grade from the final year of compulsory school and eligibility for high school up to the year 2011 through The Swedish Register of Compulsory School, Leaving Certificate, 1988–2012 [16]. The register holds information from 1988 and onwards and information on individual grades is collected at the end of each spring term for all students from all schools in Sweden, except from some schools that attend children with special needs. For the time period of the study there were three grade levels for the final grades of compulsory school; pass, pass with distinction, and pass with excellence, equaling to 10, 15, and 20 points respectively [17]. The grades levels were knowledge based, as in the student had to fulfil specified goals for each subject to reach a certain grade. The points from 16 subjects were summed up to a total grade of 0–320 points and the eligibility for high school was a passing grade in certain subjects. Through the Multi-generation register 2012 [18] and the Population education register 2004 [19] the information on participants' school grades was linked with information on the education level of the participants' mothers.

### Radiation dose reconstruction

The volume Computed Tomography Dose Index ($CTDI_{vol}$) was the metric used to evaluate the dose contributions to the center of the brain from the participants' examinations. The $CTDI_{vol}$ represents the average absorbed dose to a cylindrical polymethyl methacrylate (PMMA) phantom with a diameter of 160 mm (head) or 320 mm (body). The $CTDI_{vol,head}$ for each examination was reconstructed using the ImPACT CT Patient Dosimetry Calculator 1.0.4 (ImPACT, St. George's Healthcare NHS Trust, London, UK), using information on CT manufacturer and model, tube voltage (kV), tube current time product (mAs), collimation width and pitch for the examinations available in the PACS.

CT head scans from the original study were retrieved from the PACS at the Karolinska University Hospital in Huddinge, Stockholm. From these CT images, the effective diameters of the participants' heads were measured and compared to the diameter of the CTDI head phantom. Patients who underwent multiple examinations, such as two separate examinations of the head, two separate scans of head and face, or two overlapping series, were summed up into one within 24 hours total dose. For participants with CT scans overlapping the temporal lobe, the $CTDI_{vol,\ head}$ from each scan was accumulated, if the overlap exceeded one cm. The $CTDI_{vol,\ head}$ was used to estimate the dose to the regions of interest in the brain. Examinations where dynamic mode was used were excluded from the study, since they did not cover the center of the brain.

The CT images contained metadata in accordance with the version of the Digital Imaging and Communications in Medicine (DICOM) standard, when the examinations were made. For this reason, all the information needed to reconstruct the $CTDI_{vol,head}$ using the CT Patient

Dosimetry Calculator was not explicitly known, for all examinations. CT Manufacturer and model, kV and mAs were found for all examinations. For a number of examinations, collimation width and pitch (spiral examinations) had to be approximated by studying the relationship of other scan parameters, found in the metadata. If there was any uncertainty about the approximated collimation width, the lowest reconstructed $CTDI_{vol, head}$ alternative was used. This because we wanted to avoid any misclassification of dose where patients would be erroneously allocated to higher exposure groups, since this could lead to an underestimation of any effects of radiation.

For the examinations with erroneous or lacking metadata (e.g. pitch, collimation, kV or mAs used), the $CTDI_{vol}$ was reconstructed using information from the examination's dose report (DICOM secondary capture), if it existed. As a comparison, the $CTDI_{vol,head}$ reconstructed using derived parameters (e.g., approximated pitch) was compared to existing dose reports. The radiation dose from the CT localizer radiographs were excluded from the dose reconstruction, as this dose is low compared to the rest of the CT examination.

## Statistical methods

The present study used the intention to treat, the randomization allocation from the original study, for the main analyses. The Chi-Square Test and Student's $t$-Test, for categorical and linear variables respectively [20], were used to analyze descriptive data and perform simple analyzes. Factorial logistic, for multivariate analyzes [21], were performed to analyze high school eligibility and grades (continuous and under/over 195.63 points) on their own as well as adjusted for the variables sex (male or female), age at head injury (under/over 12.08 years old), time difference between head injury and graduation (under/over 3.90 years), GOSE score at 3 months after head injury (score 8 or score below 8), and education level of mother (up to 9 years, 9–12 years, and over 12 years). The variable mean was used as the cut-off value for categorizing the continuous variables. Sub analyzes were made for the reconstructed radiation doses (0, >0–50, >50mGy) and the categorized grade score (under/over 195.63 points) and high school eligibility. SAS 9.2, 9.3, and 9.4 (SAS Institute Inc., Cary, NC, USA, 2002–2012) were used to perform all analyzes and $p = 0.05$ was used as a cut off for statistical significance. The analysis of means had an 80% power of detecting differences in means over 12 points with the existing sample size.

## Results

### Participants

The original study had five subjects who were included on two different occasions. For these subjects the first inclusion date and randomization allocation was used. Of the original 911 subjects, 1.4% (N = 13) had an incomplete PIN and could not be followed up at Statistics Sweden, 3.6% (N = 33) were too young to have graduated (under 16 year olds without compulsory school leaving grades), 2.0% (N = 18, exclusion criteria) had graduated before their head injury, and 1.6% (N = 15) were lost to follow up (Fig 1). The participation rate for the present study was 93.1% (N = 832 of N = 893). The missing data was not further analyzed. Since there was not a choice of participating we don't have the problem of selection bias, but the generalization of the results could be limited for the youngest participants with shorter follow up times who make up the group of "not yet graduated". Of the total 832 participants, 535 were boys and 297 were girls. Their age at inclusion was 6–16 years (with a mean of 12.1 years), their age at follow up was 15–18 years (with a mean of 16.0 years), and their time between injury and follow up was 1 week up to 10 years (with a mean of 3.9 years). This study used no restriction on follow up time. To be noted is that 42 (5%) participants had a follow up time under 6

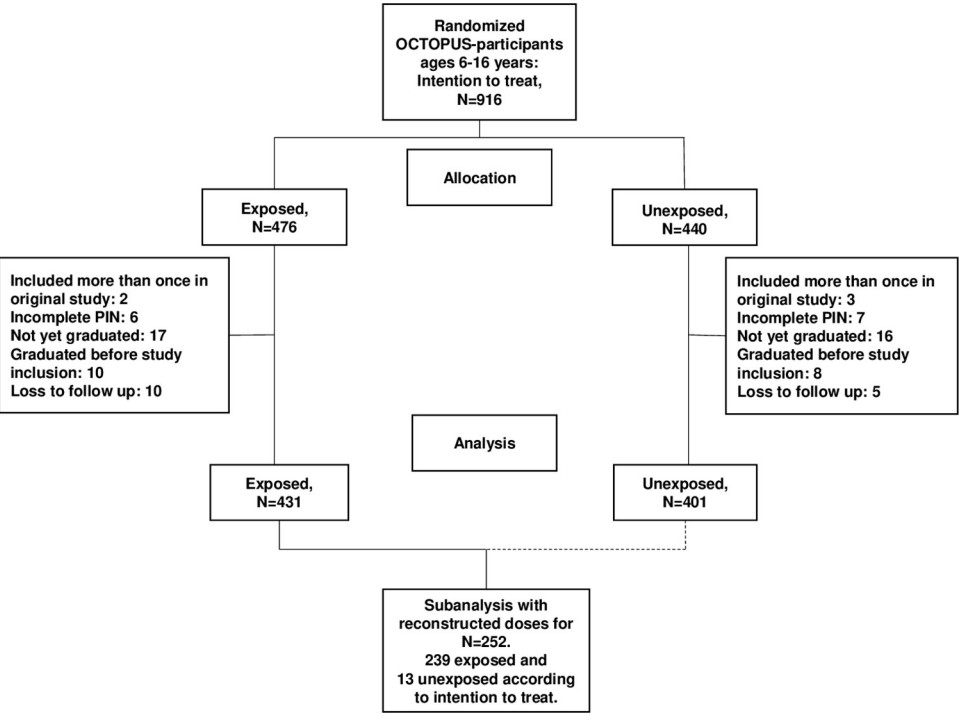

**Fig 1. Flow chart of participants.**

months. The baseline for this follow-up study is theoretically the most recent school grades at study inclusion. However, since the study is a randomized controlled trial the baseline data is not further analyzed [22, 23] and we assume that the grades are similarly distributed between the two groups, i.e. no difference in high or low grades between the exposed to ionized radiation and the non-exposed.

The exposed and unexposed participants showed no statistically significant differences in the proportion of girls and boys, age at the time of head injury, age at time of graduation, time difference between head injury and graduation, GOSE score, nor the education level of mother (Table 1). For exposed subjects, the point estimates of the proportion of subjects with high school eligibility and high school grades was up to one percent lower compared to unexposed, but neither of these differences were statistically significant (Table 2). In the ensuing analysis, the group unexposed to ionizing radiation was our reference group and the outcome of OR>1 would mean a higher probability of an unwanted outcome, in this case for the group exposed to ionizing radiation having a lower average grade or not have high school eligibility. The result OR<1 would mean a lower probability of an unwanted outcome for the exposed group, than the unexposed group. In the regression analyzes, point estimates of odds rations for no high school eligibility and lower school grade where 1.18 and 1.27, respectively, but neither of these differences were statistically significant (Table 3). The results showed no statistically significant differences between exposed and unexposed participants in their total grade score means or odds for not being in the higher grade score category (Tables 2 and 3).

## Radiation dose analyses

From the original study 289 CT head scans had been saved in the PACS at the Karolinska University Hospital in Huddinge, Stockholm. The average effective diameter of the participants' heads (N = 278) was 167±6 mm and as this was very close to the diameter of the CTDI head

**Table 1. Descriptives of participants by exposure of ionizing radiation status.**

| Variable | Exposed to ionizing radiation* | Unexposed to ionizing radiation* | p-value |
|---|---|---|---|
| Frequency (#) | 431 | 401 | - |
| Sex (% Male) | 66, N = 285 | 62, N = 250 | 0.26 |
| Age at head injury (Years, Mean (SD)) | 12.0 (2.5) | 12.1 (2.5) | 0.72 |
| Age at graduation (Years, Mean (SD)) | 16.0 (0.3) | 16.0 (0.4) | 0.30 |
| Time between head injury and graduation (Years, Mean (SD)) | 3.9 (2.4) | 3.8 (2.5) | 0.62 |
| GOSE score 3 months after head injury (% of a score of 8) | 88, N = 374 | 86, N = 340 | 0.52 |
| Mother's education level (N(%)) | | | 0.55 |
| *Max. 9 years* | 45 (11) | 52 (13) | |
| *>9–12 years* | 217 (51) | 196 (49) | |
| *Higher education* | 161 (38) | 151 (38) | |

Note 1: *According to intention to treat.

phantom, no size corrections were made to the reconstructed $CTDI_{vol, head}$ values. For 37 patients a within 24 hours total dose was calculated. Six examinations where dynamic mode had been used were excluded from the study. For 15 examinations erroneous metadata was found and $CTDI_{vol}$ was reconstructed as far as possible with PACS data and DICOM secondary exam information. For one of the examinations an assumption about the rotation time had to be made. As a comparison, the $CTDI_{vol,head}$ reconstructed using derived parameters (e.g., approximated pitch) was compared to existing dose reports (N = 9). The difference between the reconstructed $CTDI_{vol,head}$ and reported $CTDI_{vol,head}$ was 5–14%, with the reported doses being higher than the reconstructed ones. In total, the dose to the region of interest in the brain was reconstructed for 279 participants' examinations. Of these, 27 could not be matched with the data set. Of the 252 examinations, 13 were participants originally randomized to the observation group. The head size remained the same for the remaining sub group.

The dose range for the reconstructed doses was 22.2–157.4 mGy with a mean of 61.7 mGy. The comparison group comprised of the participants in the observation group with no known reassignment (N = 381). The dose range for the lower dose group was 22.2–49.4 mGy with a median of 45.2 mGy (N = 126) and the dose range for the higher dose group was 50.2–157.4 mGy with a median of 78.2 mGy (N = 126). Results for the different exposure groups, 0 mGy, >0–50 mGy, and >50 mGy, showed no statistically significant differences in their proportions of participants being in a higher scoring grade category or eligibility for high school (Table 4).

**Table 2. Simple comparisons of participants exposed and unexposed to ionizing radiation in high school eligibility, total grade score, grade in mathematics and the Swedish language, Chi-square test and students *t*-test.**

| Variable | Exposed to ionizing radiation* | Unexposed to ionizing radiation* | p-value |
|---|---|---|---|
| High school eligibility (N = 832, % Eligible) | 89%, N = 431 | 90%, N = 401 | 0.44 |
| Total grade score (N = 832, Mean (range)) | 194.5 (131.6–257.4), N = 431 | 196.8 (133.9–259.7), N = 401 | 0.59 |
| Math grade (N = 817, Mean (range)) | 11.5 (7.1–15.9), N = 426 | 12.0 (7.7–16.3), N = 391 | 0.11 |
| Swedish language grade (N = 807, Mean (range)) | 12.3 (8.0–16.6), N = 420 | 12.4 (8.2–16.6), N = 387 | 0.66 |

Note 1: *According to intention to treat.

**Table 3. Crude and multivariate comparisons of participants exposed and unexposed to ionizing radiation in not having high school eligibility and having lower grade scores.**

| Exposure* | Crude OR (95% CI) N = 832 | Multivariate OR (95% CI)** N = 832 |
|---|---|---|
| *No high school eligibility* | | |
| Unexposed to ionizing radiation | 1 | 1 |
| Exposed to ionizing radiation | 1.19 (0.76–1.86) | 1.27 (0.79–2.03) |
| *Lower grade score**** | | |
| Unexposed to ionizing radiation | 1 | 1 |
| Exposed to ionizing radiation | 1.18 (0.90–1.54) | 1.18 (0.89–1.57) |

Note 1: *According to intention to treat.

Note 2: **Adjusted for sex, age at head injury, time difference, GOSE score at 3 months after injury, and education level of mother.

Note 3: ***Lower grade score is a grade under 195.63 points.

## Discussion

In the largest study hitherto conducted on CT head examination in childhood and high school eligibility and grade scores at end of compulsory school and where radiation doses where randomly assigned, no statistically significant effects on high school eligibility or grades at the end of compulsory school were found. Furthermore, the more specific dose analyzes showed no statistically significant differences between the different dose categories and their proportions of high school eligible and higher scoring grade category.

A total of 832 subjects, representing 93% of the original cohort, were examined in the present study. The main known reasons for loss to follow up were incomplete PIN and not yet having graduated, i.e. having no school grades for analysis. The exposed and unexposed groups did not show any statistically significant differences in the descriptive variables; proportion of male participants, age at head injury, age at graduation, time between head injury and graduation, GOSE-score at 3 months after head injury, or education level of mother. Single and multivariate analyses did not show any statistically significant differences between the exposed and unexposed groups for high school eligibility or grades at end of compulsory school. Likewise, the dose analyzes did not show any statistically significant differences between exposure groups.

This study had the advantage of having radiation exposure randomized which is rare among studies of adverse effects of radiation. Other advantages were a high follow up rate, and inclusion of both males and females. Limitations of the study include the different follow up times, and that, even though the study is one of the larger of its kind, the sample size may have been too small to detect small variations between exposed and non-exposed. The small

**Table 4. Percentages of participants eligible for high school and with higher grade scores in the different dose categories, Chi-square test with 3 categories.**

| Dose to center of the brain | 0 mGy* | >0–50 mGy** | >50 mGy*** | *p*-value |
|---|---|---|---|---|
| High school eligibility | 90% | 92% | 86% | 0.23 |
| Higher grade score**** | 51% | 48% | 47% | **0.66** |

Note 1: *According to known treatment, N = 381.

Note 2: **Dose range: 22.2–49.4 mGy, Mean: 45.2 mGy, and N = 126.

Note 3: ***Dose range: 50.2–157.4 mGy, Mean: 78.2 mGy, and N = 126.

Note 4: ****Higher grade score is a grade over 195.63 points.

differences found were generally in favor of the unexposed, or of those exposed to lower doses of radiation compared to those exposed to higher doses, but neither of the results were statistically significant. Whether these differences had been statistically significant in a larger study cannot be told, but it is reassuring that any effect on school grades of computed tomography of the head is too small to be detected in a study of over 800 study subjects. Another limitation is that the present study measured only eligibility for high school attendance, and not actual high school attendance or completion. Like Blomstrand et al. [13], we found no differences between exposed and unexposed participants, nor between lower and higher doses of exposure that could have been expected from the study of Hall et al. [12]. However, these previous studies examined participants who had been exposed at a much younger age (up to 18 months) and thus in a more vulnerable developmental period of brain development. Another potential limitation of our study is that all patients did not follow the study protocol. In the original OCTOPUS-study, 96.5% received the original treatment (observation vs. exposure to computed tomography). In addition, 8.9% of the patients randomized to computed tomography were admitted to inhospital observation, and 8.6% of patients received a CT as well. Since patients with these departures from original protocol are highly likely to differ from the rest of the study population in factors such as, for example, trauma severity, injury progression and social factors that may be related to outcome. In order not to lose the unique advantages of the study's randomized design, we chose to analyze according to intention to treat [15]. The present study does not control for the treatment of the mTBI. In the original OCTOPUS-study, patients were randomized to either inhospital observation or exposure to computed tomography. It is therefore hypothetically possible that if inhospital observation were to have negative impact on ensuing school performance, any adverse effects of radiation from computed tomography may be obscured by the effects of inhospital observation. This is a problem shared for all randomized studies diagnostic procedures where randomization to a test also decides treatment [24]. Albeit shared with many other randomized trials, this is a limitation of our study. We find it unlikely, however, that one night of inhospital observation could have such a negative effect on subsequent school performance that it would threaten the validity of our results.

In conclusion, the exposure to a low ionizing radiation dose, such as from a CT examination, to the head in childhood is not associated with neither high school eligibility nor grades at end of compulsory school to such extent that it can be measured in a study of more than 800 study subjects, half of whom were randomly assigned to computed tomography.

## Acknowledgments

We acknowledge the OCTOPUS-group, especially Britt-Marie Hune and Carin Cavalli-Björkman for their support with accessing data and information from the OCTOPUS-study. Furthermore, we acknowledge Henrik R Andersson for helping us getting started with the radiation dose reconstructions.

## Author Contributions

**Conceptualization:** Elina Salonen, Robert Bujila, Jean-Luc af Geijerstam, Håkan Nyman, Olof Flodmark, Peter Aspelin, Magnus Kaijser.

**Data curation:** Elina Salonen, Robert Bujila, Jean-Luc af Geijerstam, Peter Aspelin, Magnus Kaijser.

**Formal analysis:** Elina Salonen, Robert Bujila, Magnus Kaijser.

**Funding acquisition:** Jean-Luc af Geijerstam, Håkan Nyman, Olof Flodmark, Peter Aspelin, Magnus Kaijser.

**Investigation:** Elina Salonen, Robert Bujila, Magnus Kaijser.

**Methodology:** Elina Salonen, Robert Bujila, Magnus Kaijser.

**Project administration:** Elina Salonen, Robert Bujila, Magnus Kaijser.

**Resources:** Elina Salonen, Robert Bujila, Jean-Luc af Geijerstam, Peter Aspelin, Magnus Kaijser.

**Supervision:** Elina Salonen, Robert Bujila, Magnus Kaijser.

**Visualization:** Elina Salonen.

**Writing – original draft:** Elina Salonen, Robert Bujila, Magnus Kaijser.

**Writing – review & editing:** Elina Salonen, Robert Bujila, Jean-Luc af Geijerstam, Håkan Nyman, Olof Flodmark, Peter Aspelin, Magnus Kaijser.

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
