## [Decision Letter · Decision Letter 0]

19 Dec 2022

PONE-D-22-29193Academic achievement after a CT examination toward the head in childhood: Follow up of a randomized controlled trial.PLOS ONE

Dear Dr. Salonen,

Thank you for submitting your manuscript to PLOS ONE. After careful consideration, we feel that it has merit but does not fully meet PLOS ONE’s publication criteria as it currently stands. Therefore, we invite you to submit a revised version of the manuscript that addresses the points raised during the review process.

Please make the suggested minor revisions to the manuscript below. Address each point by point. Please submit your revised manuscript by Feb 02 2023 11:59PM. If you will need more time than this to complete your revisions, please reply to this message or contact the journal office at plosone@plos.org. Please include the following items when submitting your revised manuscript:A rebuttal letter that responds to each point raised by the academic editor and reviewer(s). You should upload this letter as a separate file labeled 'Response to Reviewers'.A marked-up copy of your manuscript that highlights changes made to the original version. You should upload this as a separate file labeled 'Revised Manuscript with Track Changes'.An unmarked version of your revised paper without tracked changes. You should upload this as a separate file labeled 'Manuscript'.If applicable, we recommend that you deposit your laboratory protocols in protocols.io to enhance the reproducibility of your results. Protocols.io assigns your protocol its own identifier (DOI) so that it can be cited independently in the future. For instructions see: https://journals.plos.org/plosone/s/submission-guidelines#loc-laboratory-protocols. Additionally, PLOS ONE offers an option for publishing peer-reviewed Lab Protocol articles, which describe protocols hosted on protocols.io. Read more information on sharing protocols at https://plos.org/protocols?utm_medium=editorial-email&utm_source=authorletters&utm_campaign=protocols.

We look forward to receiving your revised manuscript.

Kind regards,

Aaron Specht

Academic Editor

PLOS ONE

Journal Requirements:

2. Please ensure that you have specified (1) whether consent was informed and (2) what type you obtained (for instance, written or verbal, and if verbal, how it was documented and witnessed). If your study included minors, state whether you obtained consent from parents or guardians. If the need for consent was waived by the ethics committee, please include this information.

Reviewers' comments:

Reviewer's Responses to Questions

**Comments to the Author**

1. Is the manuscript technically sound, and do the data support the conclusions?

Reviewer #1: Yes

Reviewer #2: Partly

Reviewer #3: Yes

2. Has the statistical analysis been performed appropriately and rigorously? 

Reviewer #1: Yes

Reviewer #2: Yes

Reviewer #3: N/A

3. Have the authors made all data underlying the findings in their manuscript fully available?

Reviewer #1: No

Reviewer #2: Yes

Reviewer #3: No

4. Is the manuscript presented in an intelligible fashion and written in standard English?

Reviewer #1: Yes

Reviewer #2: Yes

Reviewer #3: Yes

5. Review Comments to the Author

Reviewer #1: This is an interesting study assessing if ionizing radiation dose from a CT examination of the head at the age of 6-16 years affects academic performance and high school eligibility at the end of compulsory school.

As this is a follow-up analysis, in the methods section can the authors be explicit about what will be defined as baseline for analysis.

Also mention any handling of missing data - if any approach will be used.

Mention checking of model assumptions.

Table 1 - update title e.g Descriptives of participants by exposure of ..... status.

Table 1 includes footnote "intention to treat" - can authors define this in the methods section.

Table 1 headers state exposed and unexposed - for easy readability do you mean exposed to ionizing radiation - can this be reflected in the header and in methods section.

Table 2 - include ranges in your results.

Logistic regression has been stated the main method of analyses - in the methods state that unexposed is your reference and what the and OR<1 or OR>1 would mean in terms of probability of ?reduced academic performance and high school eligibility.

The results are not surprising, if you observed no differences in baseline characteristics in terms of you variables of interest, both the Crude OR and Multivariate OR. Considering the data is coming from an RCT where in general students were generally comparable only differences based on randomisation group. This should be expanded further in discussion. Why did the authors not consider also adjusting for randomisation group.

Reviewer #2: Thank you for the opportunity to review this manuscript! This is a fantastic study to address concerns of low-level radiation. There are some issues I would like to discuss and have address prior to publication.

Overall, I think the article fails to address and control for care of the subjects following their mild traumatic brain injury (mTBI). If a patient was discharged with a mTBI, what is the likely hood they followed the physician's instructions compared to the patients that were held for observation. A likely cause of the difference between exposed and non-exposed subjects was that the discharged patients did not properly follow their treatment plan resulting in improper healing. The study should include some way of controlling for the difference in the standard of care. One possible way of doing this would be comparing the results to subjects who did not experience a mTBI to see if the change in scores was consistent between exposed and non-exposed subjects.

In the “Radiation dose reconstruction” section, second to last paragraph, it is mentioned that the lowest reconstructed CTDI_vol was used. It is far more common to use the highest reconstructed dose. Using the highest dose represents the worst-case scenario that a patient could receive which is generally the main concern in the health physics field.

Additionally, in the last paragraph mentions erroneous metadata. Some more information here on what criteria was used to define erroneous metadata would be beneficial.

Thank you again! I thought you did a great job to address the topic!

Reviewer #3: For Materials and methods section, more reasoning should be included. Why is the study done in such small size population? What are the shortcomings of methodology of the study? What are the reasons behind applying the methods for this study? Questions like these need to be answered in the Materials and methods section. Also, for the statistical methods why Chi-Square Test and Student’s t-Test were performed to analyze the data must be explained in detail. For the results section I think some calculations of the statistical analysis should be included. Some graphical representation of the results might be better for understanding the results of the study.

6. PLOS authors have the option to publish the peer review history of their article (what does this mean?). If published, this will include your full peer review and any attached files.

Reviewer #1: No

Reviewer #2: **Yes: **Chandler Burgos

Reviewer #3: No

---

## [Author Response · Author response to Decision Letter 0]

5 Mar 2023

PLOS ONE

PONE-D-22-29193

Academic achievement after a CT examination toward the head in childhood: Follow up of a randomized controlled trial.

March 5, 2023

Dear Reviewers,

Thank you for very much for your corrections and comments, we sincerely believe they have enhanced the quality of the article. We have made the following changes according to your input:

 Reviewer #1: This is an interesting study assessing if ionizing radiation dose from a CT examination of the head at the age of 6-16 years affects academic performance and high school eligibility at the end of compulsory school.

1 As this is a follow-up analysis, in the methods section can the authors be explicit about what will be defined as baseline for analysis.

 Revised

 The baseline for this follow-up study is theoretically the most recent school grades at study inclusion. This information would only be available for the participants included at approximately 12 years of age, since the school system in Sweden starts with grades in grade 6. However, since the study is a randomized controlled trial the baseline data is not further analyzed (Senn, S., Seven myths of randomisation in clinical trials. Statist. Med., 2013. 32(9): p. 1439-1450 and Moher, D., et al., CONSORT 2010 Explanation and Elaboration: updated guidelines for reporting parallel group randomised trials. BMJ, 2010. 340(c869)) and we assume that the grades are similarly distributed between the two groups, i.e. no difference in high or low grades between the exposed to ionized radiation and the non-exposed. As is defined in the CONSORT statement (Moher et al., 2010) we have presented the baseline demographic and clinical characteristics in table 1. This has been made more clear in the results section.

Results, first paragraph.

2 Also mention any handling of missing data - if any approach will be used.

 Revised

 Concerning the handling of missing data, we could only have analyzed the group “not yet graduated” (Ntotal=33) for possible differences, although we know that they as a group must be younger, have participated most recently (shorter follow up time) than the participants included in the present study, but that they were in equal numbers in the allocated treatment groups. However, a possible problem of this could be that shorter follow up times could affect the results, which is a discussion for the generalizability of the study. As mentioned in the article, first inclusion date was used for the “doubles” and the ones who had “graduated before inclusion” to the original study were excluded from this follow up study. We are not able to analyze the “incomplete PIN numbers” due to missing information nor the “loss to follow up” group, since data is not traceable after linkage by Statistics Sweden. Since there was not a choice of participating, we do not have the problem of selection bias. We have added information about this to the results section.

Results, first paragraph.

3 Mention checking of model assumptions. Revised

We have added information and references for the statistical analyses in the methods section.

Methods, Statistical methods, first paragraph.

4 Table 1 - update title e.g Descriptives of participants by exposure of ..... status. Revised

5 Table 1 includes footnote "intention to treat" - can authors define this in the methods section. Revised

6 Table 1 headers state exposed and unexposed - for easy readability do you mean exposed to ionizing radiation - can this be reflected in the header and in methods section. Revised 

7 Table 2 - include ranges in your results. Revised 

8 Logistic regression has been stated the main method of analyses - in the methods state that unexposed is your reference and what the and OR<1 or OR>1 would mean in terms of probability of ?reduced academic performance and high school eligibility. Revised 

Information has been added according to suggestion.

Results, second paragraph.

9 The results are not surprising, if you observed no differences in baseline characteristics in terms of you variables of interest, both the Crude OR and Multivariate OR. Considering the data is coming from an RCT where in general students were generally comparable only differences based on randomisation group. This should be expanded further in discussion. Why did the authors not consider also adjusting for randomisation group.

We agree with the reviewer that both the rationale for our decisions about analysis and our findings can be further elaborated in the text. In the original Octopus study, 96.5% received the original treatment (observation v.s. exposure to computed tomography). In addition, 8.9% of the patients randomized to computed tomography were admitted to inhospital observation, and 8.6% of patients received a CT as well. Since these departures from original protocol are highly likely to differ from the rest of the study group in factors such as trauma severity, disease progression and social factors that may be related to outcome, we think that we may introduce a systematic error, and hence loose the advantages given by the randomization of radiation exposure, if we do not analyze the data according to intention to treat. Given the strong correlation between randomization group and exposure we abstained from introducing both these variables in the model. Information about this has been added to the discussion. 

Discussion, third paragraph.

 Revised

 Reviewer #2: Thank you for the opportunity to review this manuscript! This is a fantastic study to address concerns of low-level radiation. There are some issues I would like to discuss and have address prior to publication. 

1. Overall, I think the article fails to address and control for care of the subjects following their mild traumatic brain injury (mTBI). If a patient was discharged with a mTBI, what is the likely hood they followed the physician's instructions compared to the patients that were held for observation. A likely cause of the difference between exposed and non-exposed subjects was that the discharged patients did not properly follow their treatment plan resulting in improper healing. The study should include some way of controlling for the difference in the standard of care. One possible way of doing this would be comparing the results to subjects who did not experience a mTBI to see if the change in scores was consistent between exposed and non-exposed subjects. Revised 

We are agree with the reviewer that this may a problem. Given the design of the original OCTOPUS-study, however, it is impossible to disentangle the effects from hospital stay with the effects from CT exposure (The Evidence Base of Clinical Diagnosis: Theory and Methods of Diagnostic Research, 2nd Edition

J. André Knottnerus (Editor), Frank Buntinx (Editor)

ISBN: 978-1-405-15787-2 November 2008 BMJ Books). To remedy this, however, we address the question of whether children with mild head trauma differ from other children by comparing all the study subjects with their, supposedly, unexposed siblings. This, however, is a separate study outside of the scope of the present one, but we have added information out methodological constraints to the discussion.

Discussion, second to last paragraph

2. In the “Radiation dose reconstruction” section, second to last paragraph, it is mentioned that the lowest reconstructed CTDI_vol was used. It is far more common to use the highest reconstructed dose. Using the highest dose represents the worst-case scenario that a patient could receive which is generally the main concern in the health physics field. Revised 

In general the reconstructed CTDI(vol, head) represents the average dose, but yes, the lowest CTDI(vol, head) was used in cases where there was uncertainty about the approximated collimation width. We chose to do like this since any misclassification of dose where patients exposed to lower doses were to be allocated to higher exposure groups would risk leading to an underestimation of any negative effects. This is now explained more clearly in the text. 

Methods, Radiation dose reconstruction, third paragraph

3. Additionally, in the last paragraph mentions erroneous metadata. Some more information here on what criteria was used to define erroneous metadata would be beneficial. Revised

We tried to get all available information from the sources available to get as many doses reconstructed with the same method. With erroneous metadata we mean that the data could not be extracted from any of the sources, not that we excluded any data available. What constitutes erroneous or lacking metadata has been made more clear in the text.

Methods, Radiation dose reconstruction, last paragraph

 Reviewer #3: For Materials and methods section, more reasoning should be included. 

1. Why is the study done in such small size population? No revision

Our knowledge of the effects on cognition from ionizing radiation generally comes from much smaller studies than the present, and because of ethical reasons, randomized studies on radiation exposure are naturally rare. Therefore, the population size, although being small compared to traditional register studies, is exceptional for a randomized study in the field of radiation exposure. The advantages and limitations of the study are presented in the discussion section, the biggest shortcoming due to small population size being limitations in finding statistically significant results for possible small differences, and it is also why our conclusion of the study is not that there is no effect on cognition from low dose radiation, but if there is, it is too small to be detected “…in a study of more than 800 study subjects, half of whom were randomly assigned to computed tomography. 

2. What are the shortcomings of methodology of the study? Revised

Please see answer to Reviewer 2, question 1.

3. What are the reasons behind applying the methods for this study? Questions like these need to be answered in the Materials and methods section. Also, for the statistical methods why Chi-Square Test and Student’s t-Test were performed to analyze the data must be explained in detail.

 Revised 

The statistical methods used in the study are common within epidemiological research and describing them in detail are not within the scope of the present study. However, as mentioned above (Reviewer 1, questions 1-3), we have added information and references of the statistical methods used.

4. For the results section I think some calculations of the statistical analysis should be included. No revision.

Please see answer 3 above. 

5. Some graphical representation of the results might be better for understanding the results of the study. No revision.

 We have chosen to present the results in a similar way across the different analyses to keep the presentation coherent. Given the number of outcome variables and that we found no association between these outcome variables and exposure, we found for the whole that table-presentation was both the clearest and the most space efficient way to present our data.

Journal requirements

The new references in the article have been added to the reference list, the references in question are numbered 20-24. In addition, we have updated “Figure 1” to meet PLOS ONE’s style requirements. Furthermore, we have included the ethics statement to the first part of the Methods section.

We hope that the changes and the revised manuscript will be to your satisfaction and appropriate for publication in PLOS ONE.

Kind Regards,

Elina Salonen

---

## [Decision Letter · Decision Letter 1]

6 Apr 2023

Academic achievement after a CT examination toward the head in childhood: Follow up of a randomized controlled trial.

PONE-D-22-29193R1

Dear Dr. Salonen,

We’re pleased to inform you that your manuscript has been judged scientifically suitable for publication and will be formally accepted for publication once it meets all outstanding technical requirements.

Kind regards,

Aaron Specht

Academic Editor

PLOS ONE

Additional Editor Comments (optional):

Reviewers' comments:

Reviewer's Responses to Questions

**Comments to the Author**

1. If the authors have adequately addressed your comments raised in a previous round of review and you feel that this manuscript is now acceptable for publication, you may indicate that here to bypass the “Comments to the Author” section, enter your conflict of interest statement in the “Confidential to Editor” section, and submit your "Accept" recommendation.

Reviewer #1: All comments have been addressed

Reviewer #2: All comments have been addressed

2. Is the manuscript technically sound, and do the data support the conclusions?

Reviewer #1: Yes

Reviewer #2: Yes

3. Has the statistical analysis been performed appropriately and rigorously? 

Reviewer #1: Yes

Reviewer #2: Yes

4. Have the authors made all data underlying the findings in their manuscript fully available?

Reviewer #1: No

Reviewer #2: Yes

5. Is the manuscript presented in an intelligible fashion and written in standard English?

Reviewer #1: Yes

Reviewer #2: Yes

6. Review Comments to the Author

Reviewer #1: (No Response)

Reviewer #2: Thank you for addressing my comments and revising your paper. I enjoyed the opportunity to be apart of this review process.

7. PLOS authors have the option to publish the peer review history of their article (what does this mean?). If published, this will include your full peer review and any attached files.

Reviewer #1: No

Reviewer #2: No

---

## [Editor Report · Acceptance letter]

11 Apr 2023

PONE-D-22-29193R1 

Academic Achievement after a CT Examination toward the Head in Childhood: Follow up of a randomized controlled Trial. 

Dear Dr. Salonen:

I'm pleased to inform you that your manuscript has been deemed suitable for publication in PLOS ONE. Congratulations! Your manuscript is now with our production department. 

Kind regards, 

on behalf of

Dr. Aaron Specht 

Academic Editor

PLOS ONE